# Tetrahydrocannabinol Sensing in Complex Biofluid with Portable Raman Spectrometer Using Diatomaceous SERS Substrates

**DOI:** 10.3390/bios9040125

**Published:** 2019-10-14

**Authors:** Kundan Sivashanmugan, Yong Zhao, Alan X. Wang

**Affiliations:** 1School of Electrical Engineering and Computer Science, Oregon State University, Corvallis, OR 97331, USA; sivashak@oregonstate.edu; 2School of Electrical Engineering, The Key Laboratory of Measurement Technology and Instrumentation of Hebei Province, Yanshan University, Qinhuangdao 066004, China; zhaoyong@ysu.edu.cn

**Keywords:** tetrahydrocannabinol, portable Raman spectrometer, thin layer chromatography, surface-enhanced Raman spectroscopy, photonic crystals

## Abstract

Using thin-layer chromatography in tandem with surface-enhanced Raman spectroscopy (TLC-SERS) and tetrahydrocannabinol (THC) sensing in complex biological fluids is successfully conducted with a portable Raman spectrometer. Both THC and THC metabolites are detected from the biofluid of marijuana-users as biomarkers for identifying cannabis exposure. In this article, ultra-sensitive SERS substrates based on diatomaceous earth integrated with gold nanoparticles (Au NPs) were employed to detect trace levels of cannabis biomarkers in saliva. Strong characteristic THC and THC metabolite SERS peaks at 1601 and 1681 cm^−1^ were obtained despite the moderate interference of biological molecules native to saliva. Urine samples were also analyzed, but they required TLC separation of THC from the urine sample to eliminate the strong influence of urea and other organic molecules. TLC separation of THC from the urine was performed by porous microfluidic channel devices using diatomaceous earth as the stationary phase. The experimental results showed clear separation between urea and THC, and strong THC SERS characteristic peaks. Principal component analysis (PCA) was used to analyze the SERS spectra collected from various THC samples. The spectra in the principal component space were well clustered for each sample type and share very similar scores in the main principal component (PC1), which can serve as the benchmark for THC sensing from complex SERS spectra. Therefore, we proved that portable Raman spectrometers can enable an on-site sensing capability using diatomaceous SERS substrates to detect THC in real biological solutions. This portable THC sensing technology will play pivotal roles in forensic analysis, medical diagnosis, and public health.

## 1. Introduction

Cannabis-based drug abuse is rapidly increasing worldwide and is closely associated with violence and many other criminal activities [1,2,3,4,5]. The most prominent chemical compound in the *Cannabis sativa* L. plant is Δ^9^-tetrahydrocannabinol (THC), which is responsible for the highest degree of psychoactive activity [1,2,3,4,5]. The ability to sense a low concentration (ng/mL) of THC in biological fluids (e.g., blood, saliva, and urine) within the period of physiological activity would have significant values for potential forensic and medical applications [1,2,3,4,5]. The concentration of THC rapidly reduces with respect to time due to the metabolic breakdown of biological fluids. For instance, after smoking marijuana, the THC concentration in saliva reduces from 148 to 82.6 ng/L within 30 min. Similarly, the THC concentration in blood samples decreases from 20 to 5 ng/mL after 24 h [1,2,3,4,5,6,7].

Numerous efforts have been employed to detect THC in biofluids, including gas chromatography-mass spectrometry (GC-MS), high-performance liquid chromatography (HPLC), and enzyme-linked immunosorbent assay [3,4,8,9,10,11]. These sensing techniques usually require complex sample preparation or time-consuming, expensive procedures. Thus, there is an urgent need for an alternative, rapid, and sensitive analytical technique to accurately perceive THC in biological fluids. Surface-enhanced Raman scattering (SERS) has been increasingly adopted for chemical and biological sensing of various species at very low concentrations due to the extraordinary enhancement factors (EFs) from plasmonic nanostructures [12,13,14]. In recent years, SERS techniques have also been applied to illicit drug detection [15,16,17,18,19]. For instance, our group used hybrid photonic crystal-plasmonic SERS substrates, which were fabricated by depositing silver nanoparticles (Ag NPs) into diatom photonic crystals [13,20]. We successfully detected THC in spiked saliva samples, which achieved a detection limit of 10^−12^ M, by using a confocal Raman microscope. However, using a laboratory bench-top Raman microscope is unsuitable for on-site detection or portable sensing. In addition, although direct SERS sensing of THC from saliva is feasible, many ordinary biomolecules (e.g., protein and urea acid) or particles will disrupt direct THC SERS sensing from complex biofluid samples such as urine or blood. These biomolecules have high binding affinity to SERS substrates and induce severe interferences with strong background noise [12,21,22,23]. Therefore, it is necessary to develop a chromatography technique in tandem with SERS sensing to detect THC from real complex biofluids.

Compared with traditional GC or HPLC, thin-layer chromatography (TLC) is intrinsically suitable for portable sensing due to its exclusive advantages in a simple equipment requirement, rapid processing time, and cost effectiveness [4,8,21]. Recently, TLC in tandem with a SERS (TLC-SERS) method has been developed for various sensing applications [21,24,25,26,27,28]. Our group also demonstrated TLC-SERS sensing using diatomaceous earth substrates for food safety and illicit drug detection with ultra-high detection sensitivity [24,25,26,27]. Diatomaceous earth consists of fossilized remains of diatoms, which are hard-shelled algae [24,25,29,30]. Diatomaceous earth particles possess unique material properties in terms of high porosity, large surface area, strong adsorption capacity, and abundant hydroxyl groups at the bio-silica surface [24,25,29]. The two-dimensional (2D) periodic pores embedded within the diatom shells resemble nanoscale photonic crystal features, which can provide additional SERS EFs [24,25]. In this article, we report patterned porous diatomaceous channels as the stationary phase for separating THC from urine samples. By adding gold nanoparticles (Au NPs) onto the diatomaceous substrates, SERS sensing was conducted via a portable Raman spectrometer. Such diatomaceous TLC-SERS technique achieved 10 ppm THC sensing from spiked urine samples and clear SERS spectra from urine of volunteer marijuana users.

## 2. Materials and Methods

### 2.1. Synthesis of Gold Nanoparticles

The Au NPs were synthesized by a sodium citrate reduction method. In addition, 100 mL of aqueous solution containing 1 mM chloroauric acid was heated to a near-boiling temperature under vigorous stirring. Five mL of 1% sodium citrate solution was then added to the boiling solution and the final reaction continued for an additional hour. The final mixture solution was heated until the color of the solution became reddish-brown. The final solution was then cooled to room temperature and washed three times by deionized water to remove impurities. The Au NPs were then collected for SERS sensing.

### 2.2. Fabrication of Porous Diatomaceous TLC Plates

The diatomaceous earth powder was dried at 160 °C for 12 h in an oven and cooled to room temperature. Subsequently, 6 g of the powder was dispersed in 10 mL of 0.4% aqueous solution of carboxymethyl cellulose under vigorous stirring. The glass slides were first covered with an adhesive tape. Next, arrayed slits were cut through the tape at different widths using a sharp razor blade. Afterward, the mixed solution was deposited onto the patterned glass substrate by spin-coating at 800 rpm for 10 s and the tape was then removed. The diatomaceous plates were dried in a dark place and baked at 110 °C for 3 h to increase the adhesion to the glass surface.

### 2.3. Biofluid Samples Preparation

Saliva samples were collected directly after (i.e., within 15 min of) marijuana use from volunteers. The collected saliva samples (2 mL) were mixed with an equal amount of water and centrifuged at 8000 rpm for 15 min to remove oral impurities and food residues. Then, the saliva samples were stored at 4 °C to suppress degradation. For SERS measurements, the as-prepared saliva samples (2 µL) were measured using diatomaceous SERS substrates. The urine samples (15 mL) were collected from both non-marijuana users and marijuana users in glass vials in the morning for multiple days. The urine samples from the non-marijuana volunteers were mixed with various concentrations of THC (1000 ppm to 10 ppm) for SERS characterization. The urine samples from marijuana-users and THC-spiked urine samples were filtered with syringe filters and diluted with water for the SERS measurement. All samples were analyzed within 3 h of collection.

### 2.4. TLC-SERS Method

The TLC-SERS detection was designed to detect THC in urine samples, as shown in Figure 1a. To separate THC from the urine solution, 2 µL of the urine sample was drop-cast on the TLC plate (i.e., 12 mm from the edge of the plate) and air-dried. Next, the TLC plate with the urine sample was immersed in the development chamber with the mobile-phase eluent (chloroform-acetone-ammonium hydroxide, 40/20/40 *v*/*v*), which migrates along the porous diatomaceous channels driven by the capillary force [24,25]. After THC was separated from other species, the TLC plate was removed from the development chamber and air-dried, as shown in Figure 1a. The isolated THC was visualized using fast blue salt colorimetry. Subsequently, 2 μL of Au NPs in solution were directly dropped onto the separation spot and measured by the portable Raman spectrometer.

### 2.5. Portable Raman Spectrometer

The SERS spectra were acquired by a portable Raman spectrometer (B&W TEK, i-Raman plus, USA) as indicated in Figure 1a. Before the SERS measurement, the working distance of the portable Raman spectrometer was optimized with a Si wafer ranging from 2 to 8 mm. Figure 1b illustrates the Raman spectra of different working distances from the Raman probe tip to the Si wafer. The Raman peak of the Si at 518 cm^−1^ was used to optimize the working distance, which shows that 5-mm is the optimal working distance. This is demonstrated in Figure 1c. Subsequently, all SERS measurements were conducted at a 5-mm working distance using a 420 mW and 785 nm wavelength excitation laser. The laser spot size is 85 μm in diameter. The SERS spectra were acquired using 10-s integration time in the spectral range from 400 to 1800 cm^−1^. The SERS spectra was averaged from 20 consecutive measurements.

### 2.6. Enhancement Factor Calculation 

The EF was calculated according to the formula below.
$$EF=\;\frac{I_{SERS}\times N_{Bulk}}{I_{Bulk}\times N_{SERS}}$$
where *I_Bulk_* and I*_SERS_* are the peak intensity of the normal Raman measurement with 10^−3^ M THC solution and SERS measurement with 10^−7^ M THC solution, respectively. *N_Bulk_* and *N_SERS_* are the number of THC molecules within the laser spot of the portable Raman spectrometer for the normal Raman measurement and SERS measurement, respectively. *N_Bulk_* and *N_SERS_* was calculated using the equation below.
$$N_{Bulk}=C_{0}\times V_{0}\times N_{A}\times \;{\left(\frac{d}{D_{0}}\right)}^{2}$$
where *C*_0_ is the concentration of THC for the normal Raman measurement, *V*_0_ is the volume of THC solution for a normal Raman measurement, *N_A_* is the Avogadro constant, *d* is the laser spot area, and *D*_0_ is the diameter of the liquid spot in which the THC molecule is distributed on the glass substrate.
$$N_{SERS}=C_{SERS}\times V_{SERS}\times N_{A}\times \;{\left(\frac{d}{D_{SERS}}\right)}^{2}$$
where *C_SERS_* is the concentration of the THC for the SERS measurement, *V_SERS_* is the volume of THC solution for the SERS measurement, and *D_SERS_* is the diameter of the liquid spot in which the THC molecule is distributed on the diatomite substrate.

## 3. Results and Discussion

### 3.1. Characterization of Plasmonic Photonic Materials

Field-emission scanning electron microscopy (FE-SEM) images of diatomaceous earth on glass substrates are shown in Figure 2a. The commercial diatomaceous particles are mostly dish-shaped with 2D periodic pores, which can generate strong guided-mode resonance [24,25,27]. The diatomaceous particles’ size ranges from 10 to 25 µm. The diatomaceous film deposited on the glass substrate works as the stationary phase for the TLC plate. The Au NPs were synthesized using a simple reduction method [22]. Figure 2b depicts the ultraviolet-visible absorption spectrum of the Au NPs. The surface plasmon resonance peak was located at 521 nm with a narrow bandwidth, which suggests that the NPs were mono-dispersed in the aqueous solution without significant aggregation. The FE-SEM image of the Au NPs was presented in the inset of Figure 2b, which demonstrated an average diameter of ~60 nm and exhibited a nearly spherical shape. After TLC separation, 5 µL of the Au NP solution was drop-casted onto the THC aggregation spot on the TLC chip. The distribution of Au NPs on the surface of the diatomaceous particle is illustrated in Figure 2c.

### 3.2. SERS Analyses of THC in Methanol and Saliva Samples

Various concentrations of THC (1–1000 ppm) in methanol solution were dispensed on the diatomaceous films, and then Au NPs were drop-casted for SERS analysis. The weak Raman signals were observed from pure THC solution (1000 ppm) on the glass substrate due to the low THC Raman activity. The SERS spectra of THC at various concentrations were examined using the 785 nm portable Raman spectrometer, as demonstrated in Figure 3a. The SERS bands of THC are located at 958, 1001, 1091, 1201, 1287, 1342, 1427, 1482, 1565, and 1601 cm^−1^, and their assigned bands are enumerated in Table 1 [15,16,17,20]. The most intense SERS peak appeared at 1601 cm^−1^, which is assigned to the C-C stretching mode. The EF for Au NPs on diatomite substrate and pure Au NPs was compared using the strong SERS band at 1601 cm^−1^. Au NPs on diatomite substrate showed the highest EF (1.4 × 10^6^) compared to pure Au NPs (1.2 × 10^2^) due to the strong local field that occurred among the congregated Au NPs inside the diatomite pores.

Marijuana-users’ saliva samples were also used to detect THC using the diatomaceous SERS substrate. Figure 3b presents the THC SERS spectra of two marijuana-users’ saliva samples. Both saliva samples clearly indicated the THC SERS characteristic peak at 1601 cm^−1^, while other THC SERS peaks were not prominent. The SERS bands of THC in marijuana-users’ samples are located at 1000, 1067, 1230, 1345, 1409, 1454, 1548, 1601, and 1681 cm^−1^, and are assigned to their respective bonds in Table 1. The THC metabolite SERS peak appeared at 1681 cm^−1^, which is assigned to the O-C = O and indicate the formation of the THC metabolite [20]. In human biological fluids, the THC structure can be modified and form a hydroxyl metabolite and a carboxylic acid metabolite due to its rapid oxidization, which can be found after a few days of intake of THC [1,3,20]. It shows that, in the case of healthy volunteer saliva, the biomolecules present very little interference to the SERS spectrum and introduces acceptable background noise. Therefore, the THC in saliva is directly detected using SERS due to the absence of a significant biomolecule or particle interference.

### 3.3. THC Sensing in Urine Samples Using TLC-SERS

In the case of a complex solution such as plasma or urine samples, chromatography methods are required for separating THC from other biomolecules. The collected urine samples were directly measured using SERS, as shown in Figure 4a. The SERS peaks from urine samples were mostly related to urea Raman bands, which are enumerated in Table 1 [20,31,32,33]. Therefore, direct SERS sensing of THC in urine samples, either from artificially spiked urine or obtained from a volunteer marijuana user, was not successful due to the strong background interference from urea, which usually has strong binding affinity to the SERS substrate.

Chromatography separation of THC in the urine sample was required for practical sensing applications. Chloroform-acetone-ammonium hydroxide (40/20/40 *v*/*v*) eluent was used as the mobile-phase eluent. The widths of the TLC plate channels were 1, 2, and 3 mm, respectively, as shown in the inset in Figure 4b. After TLC separation, SERS spectra of 500 ppm THC were measured from various widths of TLC channels, as shown in Figure 4b. The SERS bands of THC are located at 998, 1001, 1153, 1250, 1342, 1366, 1448, 1482, and 1605 cm^−1^, which match the SERS spectra of pure THC solution. This is well-summarized in Table 1 [15,16,17,20]. Nonetheless, the strongest SERS characteristic peak is located at 1605 cm^−1^, which is assigned to the C-C stretching modes. The smallest width (1 mm) TLC channel produced the strongest SERS signals when compared to the other channels due to the superior THC confinement. Therefore, all TLC-SERS sensing measurements in this study were based on a 1-mm channel.

The resulting clear separation of THC in the THC-spiked urine sample is illustrated in the inset in Figure 4c. A quick migration of the THC on the diatomaceous earth TLC plate was observed and can be attributed to the molecular polarity of THC. The obtained retention factor (Rf), which is calculated as the ratio of the distance migrated by the target analyte to the distance migrated by the eluent solvent, was 0.69 and 0.72 cm for the concentrated urea and THC, respectively, which was shown in the inset of Figure 4c. The SERS spectra at two marked positions demonstrated strong SERS characteristic peaks of urea (1) and THC (2), as illustrated in Figure 4c. These results confirmed the THC separation from urine samples using the diatomaceous TLC plate. Next, THC at various concentrations (i.e., 1000 ppm to 10 ppm) was artificially added to urine solutions. Urine samples were collected from non-marijuana using volunteers in the morning. After the separation of THC from these artificially-contaminated urine samples, SERS spectra were measured on the THC spots, according to the Rf value. The strongest SERS peak of THC appeared at 1605 cm^−1^, as shown in Figure 4d. There are several clear urea Raman peaks at 627, 727, 780, 830, 903, 937, 1000, 1038, 1256, 1293, 1382, and 1407 cm^−1^ and THC Raman peaks at 722, 785, 932, 1164, 1210, 1367, 1392, 1478, 1582, and 1605 cm^−1^. Their assigned bands are listed in Table 1. Compared with direct THC SERS sensing from saliva samples, more THC-related Raman peaks were obtained using TLC separation. This indicated a clear separation of THC from complex samples.

To demonstrate THC sensing from real biological fluid, TLC-SERS sensing from marijuana-user’s urine samples was conducted. Before TLC separation, only urea SERS peaks can be measured and the 1165, 1245, and 1407 cm^−1^ Raman peaks of urea were clearly observed, as illustrated in Figure 5. After TLC separation, the SERS peak of THC at 1003 and 1603 cm^−1^ can be clearly observed. However, relatively few THC-related Raman peaks were obtained when compared with THC-spiked urine samples. This reflects the strong impact of the marijuana user’s urine background from other biomolecules interference.

In order to quantitatively evaluate the effectiveness of our portable SERS sensing technique, we used Principal component analysis (PCA), which is a multivariate statistical method for feature extraction, to analyze the samples we measured in our previous studies. PCA was performed using math-works (MATLAB) software based on 20 spectra for each sample type. PCA was applied to six groups of data: SERS spectra without TLC (i.e., pure THC, healthy people’s saliva and marijuana user’s saliva) and with TLC separation (i.e., urine sample, THC-spiked urine sample, and marijuana user’s urine sample), as shown in Figure 6. In this analysis, the first three PCs of the SERS spectra account for PC1 (98.25%), PC2 (0.75%), and PC3 (0.50%) of the variance for the six groups of samples, as displayed in Figure 6. It is very clear that PC1 is the main principle component and should account for the major score. When comparing pure THC (200 ppm) and marijuana users’ saliva samples using the SERS-only sensing method, the PCA plot shows highly congregated clusters arranged in close proximity to each other. The PCA results indicated that saliva samples contained less molecular interference and the results closely matched pure THC solution. After TLC separation, each sample matrix formed a good cluster but slightly spread out along a single plane. The samples containing THC have a similar PC1 score compared to pure THC solution, while the urine samples and healthy people’s saliva without THC are far away from other samples and have a very different PC1 score. This indicates the significant difference of the SERS spectra due to the presence of THC in different biological samples. When comparing THC-spiked urine samples and marijuana user’s urine samples, the latter are further away from the pure THC, which suggests stronger background interference and matches our previous analysis. Therefore, using the PC1 score from the PCA results can effectively evaluate the presence of THC in complex biological samples.

## 4. Conclusions

In this article, we presented a THC-sensing technique using a portable Raman spectrometer to detect THC in the saliva and urine samples of marijuana users. With the high-performance diatomaceous SERS substrates, we obtained the SERS spectra of THC in saliva without pre-treating the samples. To sense THC from more complex biofluids, we developed a TLC-SERS sensing protocol to separate and detect THC molecules from urine samples of marijuana users based on porous diatomaceous microfluidic channels as the stationary phase. The described THC detection method allows for rapid and sensitive detection of THC analyte in spiked urine with concentrations ranging from 1000 to 10 ppm. Detection of THC in the marijuana user’s urine sample was clearly obtained after TLC separation. Furthermore, PCA was used to analyze the SERS spectra of all samples and shows a clear difference of THC from the complex background. In summary, such a portable SERS-sensing technique can play a pivotal role in future forensic and medical applications.

## Figures and Tables

**Figure 1 biosensors-09-00125-f001:**
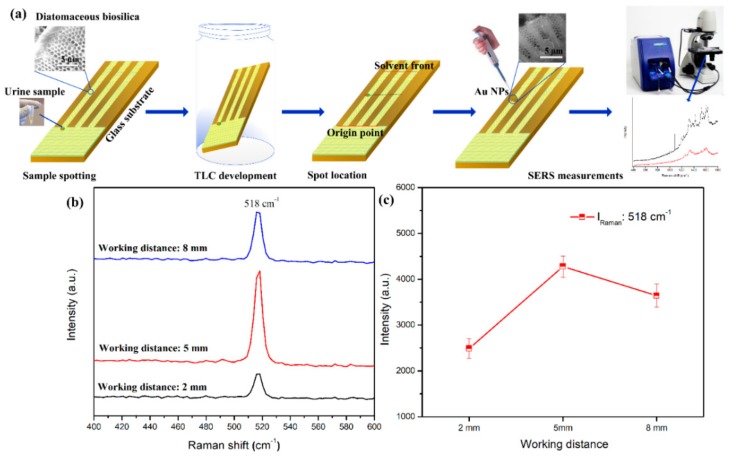
(**a**) Process of TLC-SERS detection of THC from urine samples via a portable Raman spectrometer. (**b**) The Raman spectra of a Si wafer at various working distances from the fiber tip, and (**c**) intensities versus different working distances using the I_Raman_ peak at 518 cm^−1^.

**Figure 2 biosensors-09-00125-f002:**
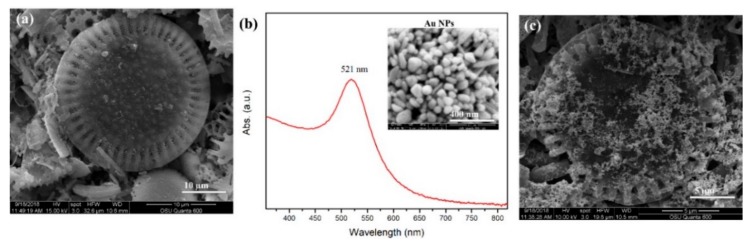
(**a**) The FE-SEM top-view image of diatomaceous particles. (**b**) The SPR spectra of as-prepared Au NPs with an inset FE-SEM image of Au NPs, and (**c**) the FE-SEM top-view image of Au NPs on the diatomaceous particle.

**Figure 3 biosensors-09-00125-f003:**
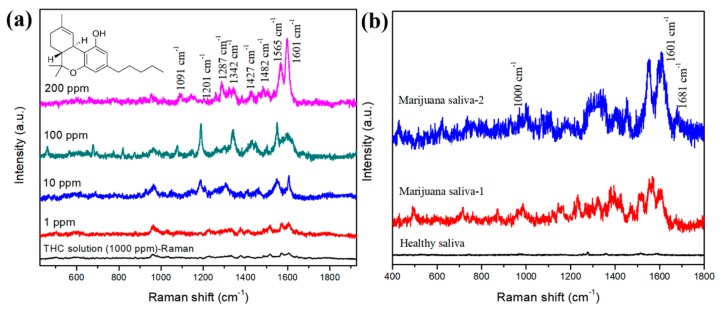
(**a**) SERS spectra of THC molecules (200, 100, 10, and 1 ppm) in methanol, and (**b**) SERS spectra from marijuana-users’ saliva samples.

**Figure 4 biosensors-09-00125-f004:**
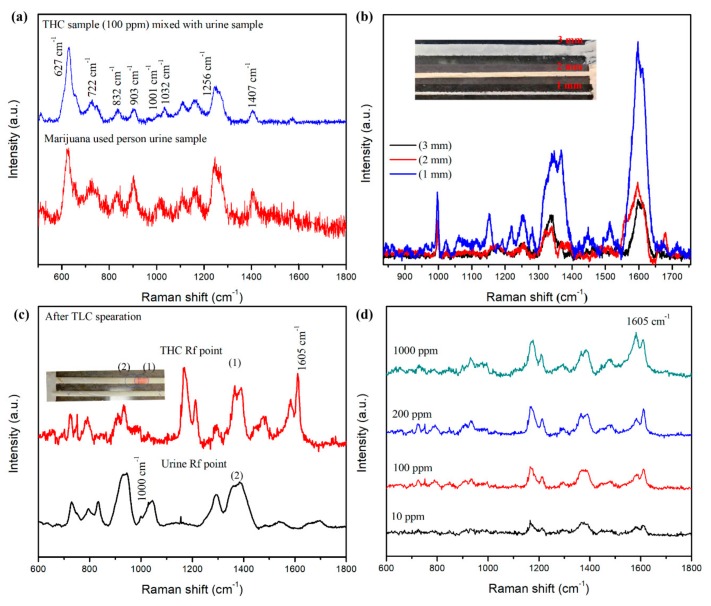
(**a**) SERS spectra of THC-spiked (100 ppm) urine, and marijuana-user’s urine samples measured from the diatomaceous SERS substrates. (**b**) TLC-SERS spectra of THC (500 ppm) from diatomaceous earth TLC plates with 1, 2, and 3-mm channel widths (inset figure shows TLC channels). (**c**) TLC-SERS spectra of THC and urea from THC-spiked urine sample, and (**d**) TLC-SERS spectra of THC from various concentrations of THC-spiked urine samples.

**Figure 5 biosensors-09-00125-f005:**
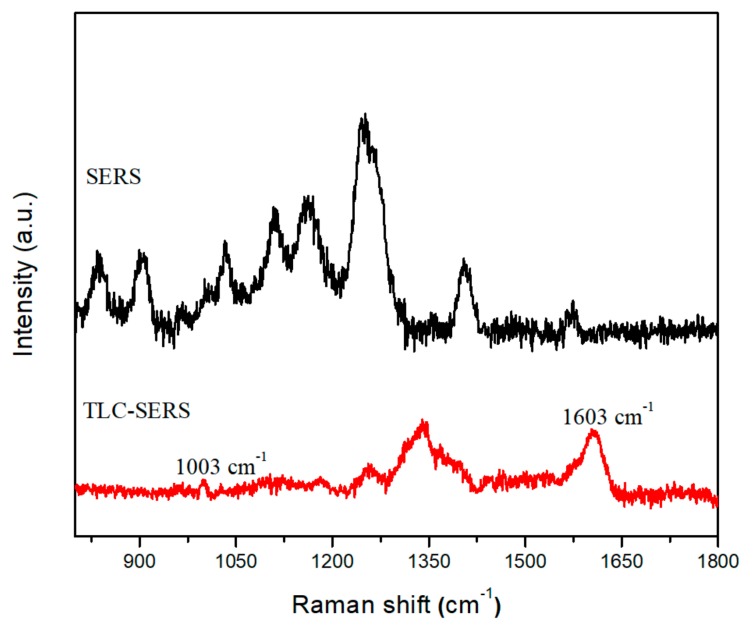
SERS spectra of THC in a marijuana-user urine sample before and after TLC separation.

**Figure 6 biosensors-09-00125-f006:**
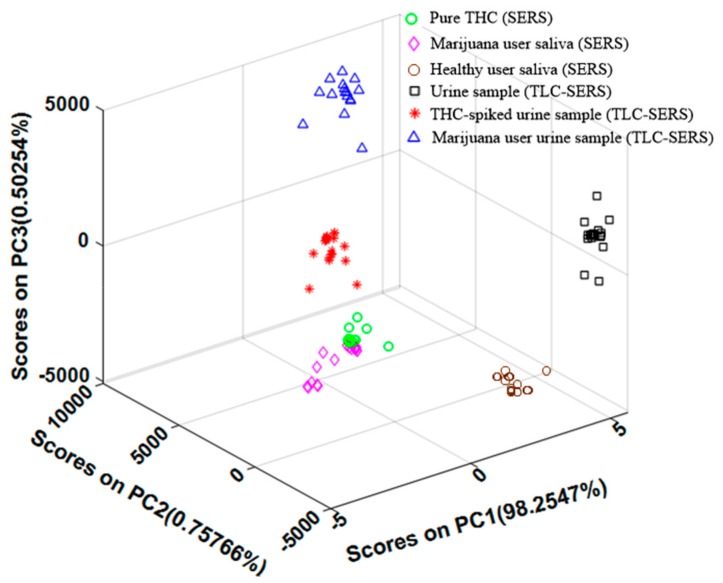
PCA scatter plot of SERS spectra of the first three PC score: pure THC (SERS), marijuana user saliva (SERS), healthy people saliva, urine sample (TLC-SERS), THC-spiked urine sample (TLC-SERS), and marijuana user’s urine sample (TLC-SERS).

**Table 1 biosensors-09-00125-t001:** Assignment of Raman peaks for urea and THC molecules.

Raman Shifts (cm^−1^)		Raman Bands Assignments
Urea	THC	
627,727		C-H bending vibration
780		C-C stretching vibration
830		C-C stretching vibration
903,937	932,958	C-O-H stretching vibration
1000,1038		N-C-N stretching vibration
	998,1001	C-C stretching vibration
	1091	C-C stretching vibration
	1153,1164,1171,1192	C-C stretching vibration
	1201,1210,1250	CH deformation
1245,1256,1293		H-C-H bending vibration
	1343,1366,1367	C-H deformation
1382,1407	1384,1392	H-C-H bending vibration
	1448,1475,1512	-C-H deformation
	1582,1584	C-C stretching
	1600–1605	C-C stretching
	1681	O-C = O stretching

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
