# Peer review of "Tetrahydrocannabinol Sensing in Complex Biofluid with Portable Raman Spectrometer Using Diatomaceous SERS Substrates"

_biosensors, 2019, doi:10.3390/bios9040125_

Round 1

Reviewer 1 Report

The manuscript, entitled "Tetrahydrocannabinol Sensing in Complex Biofluid with Portable Raman Spectrometer using Diatomaceous SERS Substrates", is written in a sufficiently clear and fluid way, both in the introduction and in the presentation of the methods and results.

My personal comments to the manuscript are shown as follows:

Referring to Figure 3, it may also be useful to include the pure THC Raman spectrum, in order to have more complete information of the analyzed molecule. Which is the Enhancement Factor (EF) of your SERS substrate? Is it comparable with other similar SERS substrate present in scientific literature? Referring to Figure 6, it would also be useful to add the data relating to "healthy saliva sample" to PCA plot.

Author Response

Thanks for the suggestion. Figure 3 (a) is updated in the revised manuscript with Raman spectra of 1000ppm THC. Also, the EF comparison of SERS substrates are added into the revised manuscript. The details are included.  Figure 6 is updated with the PCA plot of healthy saliva sample.

Page 04, Line 137-151:

The EF was calculated according to the formula:

where IBulk and ISERS are the peak intensity of the normal Raman measurement with 10-3 M THC solution and SERS measurement with 10-7 M THC solution, respectively; NBulk and NSERS are the number of THC molecules within the laser spot of the portable Raman spectrometer for the normal Raman measurement and SERS measurement, respectively. NBulk and NSERS was calculated as:

where C0 is the concentration of the THC for the normal Raman measurement, V0 is the volume of THC solution for normal Raman measurement,  NA is Avogadro constant, d is the laser spot area, and D0 is the diameter of the liquid spot in which the THC molecule distributed on the glass substrate.

where CSERS is the concentration of the THC for the SERS measurement, VSERS is the volume of THC solution for SERS measurement, and DSERS is the diameter of the liquid spot in which THC molecule distributed on the diatomite substrate.

Page 05-06, Line 175-177:

The weak Raman signals were observed from pure THC solution (1000 ppm) on the glass substrate due to the low THC Raman activity.

Page 06, Line 181-184:

The EF for Au NPs on diatomite substrate and pure Au NPs was compared using the strong SERS band at 1601 cm-1. Au NPs on diatomite substrate showed the highest EF (1.4 x 106) compared to pure Au NPs (1.2 x102) due to the strong local field occurred among the congregated Au NPs inside the diatomite pores.

Reviewer 2 Report

I recommend accepting this manuscript for publication in Biosensors with minor revision.

The preparation method of nano gold should be added in this paper。 The Ultraviolet spectra and scanning electron microscopy of gold nanoparticles are required. If possible, the authors could demonstrate the limits and reproducibility of this method for detecting THC.

Author Response

Thanks for the suggestion. Synthesis of Au NPs is added into the revised manuscript on Page 03, Line 83-89:

The Au NPs were synthesized by sodium citrate reduction method. Briefly, 100 mL of aqueous solution containing 1 mM chloroauric acid was heated to near-boiling temperature under vigorous stirring. 5 mL of 1% sodium citrate solution was then added to the boiling solution and the final reaction continued for an additional hour. The final mixture solution was heated until the color of the solution became reddish-brown. The final solution was then cooled to room temperature and washed three times by DI water to remove impurities. The Au NPs were then collected for SERS sensing.

SPR spectra and SEM images are already included in Figure 2(b).  The limit of detection was achieved by using 1 ppm THC in urine sample (spiked samples) and each measurement was based on the average data from 20 SERS spectra. The details are discussed in Section 3.3.

Reviewer 3 Report

This manuscript by Wang et al. describes a THC sensing technique using a portable Raman spectrometer to detect THC in saliva and urine samples of marijuana users. Based on a diatomaceous SERS substrates, they obtained the SERS spectra of THC in saliva without pre-treating the samples. They also applied a interesting TLC-SERS sensing protocol to separate and detect THC molecules from urine samples of marijuana users based on porous diatomaceous microfluidic channels as the stationary phase. The combination of an easy method and its use for THC detection in saliva and urine patients is appropriate in scope and significance for Biosensors. However, some  points should be addressed:

1) They should provide more details on the experimental part to provide the scientific community enough information for reproducing the test: (quanty of saliva used in the test, rpm centrifuge, time...)

2) The author should prove the selectivity of THC SERS signal  in presence of common substances on saliva or describe a protocol to avoid a possible false positive 

Author Response

(Q#1) They should provide more details on the experimental part to provide the scientific community enough information for reproducing the test: (quanty of saliva used in the test, rpm centrifuge, time...).

(R#1) Thanks for the suggestion. The preparation of saliva samples is added into the revised manuscript on Page 03, Line 100-103:

The collected saliva samples (2 mL) were mixed with an equal amount of water and centrifuged at 8000 rpm for 15 minutes to remove oral impurities and food residues. Then, the saliva samples were stored at 4°C to suppress degradation. For SERS measurement, the as-prepared saliva samples (2 µL) were then measured using diatomaceous SERS substrates.

(Q#2) The author should prove the selectivity of THC SERS signal in presence of common substances on saliva or describe a protocol to avoid a possible false positive.

(R#2) The centrifuged method significantly reduced impurities in saliva samples. Generally, the biomolecules present in saliva samples induces minimum level interference to the THC SERS measurement.